# The *Lactobacillus* as a Probiotic: Focusing on Liver Diseases

**DOI:** 10.3390/microorganisms10020288

**Published:** 2022-01-26

**Authors:** Jin-Ju Jeong, Hee Jin Park, Min Gi Cha, Eunju Park, Sung-Min Won, Raja Ganesan, Haripriya Gupta, Yoseph Asmelash Gebru, Satya Priya Sharma, Su Been Lee, Goo Hyun Kwon, Min Kyo Jeong, Byeong Hyun Min, Ji Ye Hyun, Jung A Eom, Sang Jun Yoon, Mi Ran Choi, Dong Joon Kim, Ki Tae Suk

**Affiliations:** Institute for Liver and Digestive Diseases, Hallym University College of Medicine, Chuncheon 24252, Korea; jj_jeong@hallym.ac.kr (J.-J.J.); heejin773@gmail.com (H.J.P.); qjarlf987@naver.com (M.G.C.); epark312@hallym.ac.kr (E.P.); lionbanana@hallym.ac.kr (S.-M.W.); vraja.ganesan@gmail.com (R.G.); phr.haripriya13@gmail.com (H.G.); yagebru@gmail.com (Y.A.G.); satyapriya83@gmail.com (S.P.S.); qlstn5549@gmail.com (S.B.L.); ninetjd@naver.com (G.H.K.); astella525@gmail.com (M.K.J.); wooju7023@gmail.com (B.H.M.); jiy25n@naver.com (J.Y.H.); eomjunga32@naver.com (J.A.E.); ysjtlhuman@gmail.com (S.J.Y.); choimi316@naver.com (M.R.C.); djkim@hallym.ac.kr (D.J.K.)

**Keywords:** probiotics, liver disease, *Lactobacillus*

## Abstract

Over the past decade, scientific evidence for the properties, functions, and beneficial effects of probiotics for humans has continued to accumulate. Interest in the use of probiotics for humans has increased tremendously. Among various microorganisms, probiotics using bacteria have been widely studied and commercialized, and, among them, *Lactobacillus* is representative. This genus contains about 300 species of bacteria (recently differentiated into 23 genera) and countless strains have been reported. They improved a wide range of diseases including liver disease, gastrointestinal diseases, respiratory diseases, and autoimmune diseases. Here, we intend to discuss in depth the genus *Lactobacillus* as a representative probiotic for chronic liver diseases.

## 1. Introduction

The definition of probiotics has been changed constantly. The most recent definition is “live microorganisms, which when consumed in adequate amounts, confer a health effect on the host” [1]. Because of the properties of probiotics, many studies have been conducted on their effects on various diseases. For the safety of using probiotics, various guidelines have been established, including determining antibiotic resistance/susceptibility patterns [2]. Probiotics are already being used to treat or prevent human diseases, conditions, and syndromes. They have also been shown to have a positive effect on neuroinflammation and pain, as well as seasonal disease infections. They have a multifaceted effect, including a protective role in the gut. They compete with pathogens to produce direct antimicrobial effects and indirectly enhance intestinal barrier function [3]. They also modulate the host’s local and systemic mucosal immune systems and induce inhibitors of proinflammatory cytokine production [4,5]. Many mechanisms are affected depending on the strain specificity, even in the same species [6].

Many *Lactobacillus* have a long history of use in food. This is because *Lactobacillus* is a lactic acid-producing bacterium and has “generally recognized as safe” status. Currently, there is growing interest in its use as a dietary supplement for humans and animals [7]. Bacteria belonging to the genus are found in the oral cavity, intestines, and vagina [8,9,10]. *Lactobacillus* spp. could improve conditions such as gastrointestinal diseases, allergies, and liver disease through various mechanisms, such as producing metabolites that can directly inhibit pathogens, exhibiting immunomodulatory effects, and changing the intestinal microbiota [5,11,12,13]. Microorganisms in the gut are also known to affect liver disease. This is because the venous system of the portal circulation defines the gut−liver axis, and there is a close anatomical and functional interaction between the gastrointestinal tract and the liver [14]. The disease-improving effects of various probiotics have been confirmed in non-alcoholic and alcoholic liver disease.

This review will highlight the effects *Lactobacillus* has on various diseases, especially liver diseases. Finally, as probiotics, we provide insight into how *Lactobacillus spp.* works in various diseases, particularly liver disease.

## 2. Probiotics

Gut microbiome alteration using fecal material is an ancient practice. However, the application of specific strain-based methods is only five decades old. In 1965, the first probiotics definition emerged, which only referred to the bacterial products at that time and those known to promote the growth of other groups of bacteria [15]. Later, in 1989, living microbes were also included in probiotics; however, they were only linked with nutritional health. The newest definition of probiotics includes living microbes that can be ingested and produce beneficial health effects that are not only limited to nutritional outcomes. However, all of the above definitions indicate that probiotics produce beneficial health effects through various mechanisms such as improving the eubiosis, alleviating intestinal health, and strengthening the immune system [16]. Initially, probiotics were limited to *Lactobacillus* and *Saccharomyces* genera and presented positive preventive outcomes against *Clostridium difficile* infections [17].

The human microbiota, known for their intra-site changeable relationship, for instance parietal microbiota (microbes living in mucus layer and/or in intestinal wall), are closely related with luminal microbiota (microbes living in digested food and/or transit stool). Interestingly, microbiota composition is very dynamic and person specific, which can be influenced by diet, probiotic intake, intestinal environment, and other host-dependent factors that create some transiently new bacterial stains [18].

The close relationship between the microbiota and immune system deepens the understanding about microbial component involvement in energy homeostasis and glucose and lipid metabolism [19]. It is identified as a key regulatory mechanism that has is involved in the establishment and progression of various metabolic diseases by compromising the gut barrier function through changing the gut microbiota composition [16]. Moreover, particular types of alteration in the gut microbiota composition can lead to higher T cells accumulation in the gut of high-fat diet consuming obese individuals, and increases the obesity dependent mortality rate [20].

In addition, the gut microbiota is crucial for the regulation of cognitive functions. Multiple numbers of human and animal trials represent the pivotal role of the gut microbiota in the development of cognitive functions, regulation of emotions, and in making the person to person communication by powering the neuronal-immune system, which can help in neuronal cell differentiation, synaptic plasticity, and axonal development [21].

In contrast, gut microbial compositional impairment has been related to numerous psychiatric disorders like depression, autism, alcohol-related encephalopathy, and other disorders. For example, the depressed patients’ gut microbiota is less diverse, which can possibly help to increase the proinflammatory status and cortisol level, and alter the tryptophan metabolism. Moreover, the fecal microbiota transplantation from these depressed patients into the microbiota depleted animal model mimics the depression associated with pathophysiological and behavioral characteristics similar to patients [22]. These interesting findings about a close association between the gut and brain open a new avenue of probiotics-based psychopathological intervention via modulating the gut composition. In 1910, an improvement in depression related psychopathology was observed with the supplementation of *Lactobacillus* [16]. Likewise, supplementation with *L. fermentum* and/or *L. plantarum* also showed positive and beneficial health effects in hospitalized patients, and reduced the colonization of nosocomial multi-drug resistance bacterial strains such as *Pseudomonas aeruginosa*, *Acinetobacter baumannii*, or *Candida albicans* [23,24].

Surprisingly, probiotic’s health boosting effects are not limited to their strain level, they are extended to their metabolites; for an example, supernatant collected from probiotics liquid culture are also capable of limiting the growth as well as resistance gene transmission of carbapenemase-producing and extended-spectrum *β*-lactamase-carrying *Enterobacteriaceae* [25]. The microbiota act in a multidirectional manner at the same time by limiting the expression of the virulence factor-related genes and enhancing the expression level of commensalism associated genes. These multidirectional functionalities of the microbiota can be regulated by various mechanisms: microbiota produced bioactive molecules, antiaging, boosting the immune system, by influencing the adnexal development, strengthening the sensory functions, etc. [26]. Additionally, local skin application of probiotics bacterial strains *L. acidophilus*, *L. bulgaricus,* and/or *L. plantarum* improve skin health and reduce acne through controlling the skin colonization of *Cutibacterium* acnes [27,28]. In addition, the gut microbiota has a positive influence on drug metabolism, minimizing therapeutic side effects and hepatic health [29]. The lysosomal enzyme *β*-glucuronidases released by *Bacteroides vulgatus*, *Escherichia coli*, and *Clostridium ramosum* re-activate irinotecan from its inactive state as glucuronide, which is excreted via the bile duct with bile acid directly into the gastrointestinal tract in its toxic form, and is able to cause severe digestive damage [30,31]. The mechanisms of these various probiotics have been demonstrated through preclinical and clinical trials, and among them, *Lactobacillus* is one of the most actively studied among probiotics.

## 3. Mechanisms and Applications of *Lactobacillus* as a Probiotic

There are 315 species belonging to the genus *Lactobacillus* (recently reclassified into 23 genera), and many of these bacteria have been reported as probiotics. *L. acidophilus*, *L. casei*, *L. johnsonii*, *L. reuteri*, and *L. rhamnosus,* which belong to *Lactobacillus,* are actively studied as probiotics. Most of them are resistant to gastric acid and have good adhesion to intestinal cells. For this reason, they have been applied to modulate many diseases, including gastrointestinal diseases (Table 1).

### 3.1. Mode of Action as a Probiotic

The disease alleviation effect was confirmed by preclinical and clinical trials using various species and strains belonging to the genus *Lactobacillus*. As they are effective in various diseases, various mechanisms are being elucidated. Representatively, antimicrobial activity, immunomodulatory effects, microbiota modulation, metabolites, and antitumor activity have been noted as their mechanisms of action (Figure 1).

Many *Lactobacillus* strains inhibit the growth of pathogens with antimicrobial activity [60,67]. *L. paracasei* 28.4, *L. fermentum* 20.4, and *L. rhamnosus* showed antimicrobial activity against *C. albicans*, an opportunistic pathogenic yeast. When these *Lactobacillus* were co-incubated with *C. albicans*, the mycelial growth of *C. albicans* was delayed and biofilm formation was inhibited. In addition, the expression of biofilm-specific genes was reduced in the pathogen [60]. In addition, 12 *Lactobacillus* strains exhibited an antagonistic activity against pathogenic microorganisms, *C. albicans* (ATCC 44831), *Enterococcus faecium* (ATCC 51558), *Enterobacter cloacae*, *E. coli* (ATCC 29181), *Helicobacter pylori* (ATCC 43579), *Listeria monocytogenes*, *Propionibacterium acnes* (ATCC 6919), *Shigella sonnei* (ATCC 25931), *Staphylococcus epidermidis* (ATCC 12228), and *Vibrio parahaemolyticus*, and the activity was strain dependent [67]. This inhibitory effect on pathogenic microorganisms is related to various bacteriocins, through metabolites produced by *Lactobacillus* spp. [68,69].

*Lactobacillus* could also improve the disease through immunomodulatory effects. *L. rhamnosus* ATCC 53103 increased the respiratory burst activity of blood cells. In addition, this strain significantly increased the serum-mediated killing of *E. coli* and serum immunoglobulin levels [70]. In mice, *L. acidophilus* LAFTI L10 and *L. paracasei* LAFTI L26 increased the number of immunoglobulin A (IgA), interleukin-10 (IL-10), and interferon-gamma (IFN-γ) cytokine producing cells in the small intestine. In addition, the secretion of anti-inflammatory cytokines (IL-10) and proinflammatory cytokines (IFN-γ) was increased in the systemic immune response [71]. Similarly, in a mouse peanut allergy model, *L. salivarius* HMI001 and *L. casei* Shirota (LCS) showed a partial protective effect with a high IL-10/IL-12 ratio and with high IFN-γ and IL-12, respectively [40].

In human study, *L. paracasei* F19 significantly increased its own population while decreasing the population of the pathogen *Clostridium difficile* in fecal samples [32]. This supported that *Lactobacillus* could modulate the microbiota; therefore, microbiota modulation by probiotics is being studied as one of the mechanisms to alleviate various diseases. *Lactobacillus* attenuated the progression of obesity-induced non-alcoholic fatty liver disease by regulating the microbiome [72]. In addition, the vaginal microbiota of nonpregnant sexually active women with diagnoses of bacterial vaginosis receiving *L. rhamnosus* BMX 54 was different from that of the untreated group. In the receiving BMX 54 group, the recurrence rate of bacterial vaginosis was also significantly reduced [50].

The immune response could also lead to antitumor activity. When *L. casei* was orally administered, human papillomavirus type 16 E7 protein (HPV 16 E7) specific serum IgG and mucosal IgA production were enhanced. Systemic and local cellular immunity increased, as demonstrated by the increased IFN-γ in the supernatants of vaginal lymphocytes and increased counts of IFN-γ and lymphocyte-secreting cells among splenocytes. In an E7-based mouse tumor model, *L. casei* reduced the tumor size and increased survival [73]. A novel exopolysaccharide (EPS) from *L. plantarum* 70810, a metabolite, also exhibited antitumor activity. In vitro, cell bound EPS, a novel EPS of this strain, inhibited the proliferation of HepG-2, BHC-823, and HT-29 tumor cells [65].

As shown here, disease inhibition by *Lactobacillus* spp. is not limited to one mechanism, but is highly likely to occur due to a series of actions of various mechanisms. In addition, their disease alleviating effect and related mechanisms might vary depending on the bacterial strains, not just the bacterial species level.

### 3.2. Properties as a Probiotic against Various Diseases

*Lactobacillus* is applicable to a wide range of diseases, such as gastrointestinal diseases, allergies, respiratory diseases, neurological and psychiatric diseases, liver diseases, genito-urinary infections, metabolic syndrome, cardiovascular diseases, obesity, cancer, oral disease, and vaccine adjuvants (Table 1).

There are some cases in which *Lactobacillus* has been applied to treat gastrointestinal diseases. *Lactobacillus* can reduce the abundance of the gastrointestinal microbiota. *L. paracasei* F19 was administered to children and elderly subjects twice a day for 12 weeks. The population of *C. difficile* was decreased in the fecal samples of the treated subjects [32]. *C. difficile* is a bacterium that produces toxins that cause diarrhea and enteritis [74]. Similarly, in 559 patients with acute watery diarrhea (AWD), the frequency and duration of diarrhea in the group receiving *L. rhamnosus* GG (LGG) along with an oral rehydration solution were significantly reduced compared with those in the control group receiving oral rehydration solution only [33]. This strain is safe and effective for maintaining remission in ulcerative colitis as well as in AWD [34].

In clinical practice, other bacteria are used together with *Lactobacillus*. *Bifidobacterium lactis* Bi-07 along with *L. acidophilus* NCFM was administered to 60 patients with functional bowel disorders (probiotics, *n* = 31; placebo, *n* = 29). Significantly improved bloating severity was observed in patients treated with the probiotics compared with the placebo controls [35]. The probiotic VSL#3, a mixture of more diverse bacteria, including *L. acidophilus*, *L. casei*, *L. bulgaricus*, and *L. plantarum*, alleviated dextran sulfate sodium-induced colitis in weanling rats. VSL#3 also contained *Streptococcus thermophilus* and three *Bifidobacterium* species (e.g., *longum*, *infantis*, and *breve*) [36]. Moreover, VSL#3 reduced both systemic and local anaphylaxis induced by the sensitizing allergen shrimp tropomyosin through oral treatment [37].

Other *Lactobacillus* species have also been shown to alleviate various allergy-related diseases, including atopic dermatitis, rhinitis, and food allergies. Probiotic treatment using *L. salivarius* LS01 was performed for children aged 0 to 11 years with atopic dermatitis (*n* = 43). Patients receiving this treatment had significant improvements in atopic dermatitis (SCORAD) scores and itch values from the baseline clinical parameters [38]. In patients with perennial allergic rhinitis (*n* = 49) who received milk fermented with *L. acidophilus* strain L-92, their nasal symptom-medication score was significantly improved compared to the placebo group (*n* = 24). In addition, swelling and color scores of nasal mucosae were clearly decreased [39]. Similar results were observed in Japanese cedar pollinosis patients who received *L. paracasei* strain KW3110 daily (*n* = 69). Japanese cedar pollinosis is known to be an important cause of allergic rhinitis. Patients who received KW3110 daily had significantly decreased nasal symptoms and serum levels of eosinophil cationic protein, and also increased quality of life scores compared with the placebo controls (*n* = 69). However, it was only effective when pollen scattering was low, and the effect was limited during the peak period of pollen scattering [13]. In addition, *Lactobacillus* alleviated food-induced allergies. Its prophylactic potential was investigated in a peanut sensitization model using *L. salivarius* HMI001 and LCS strains. Both strains showed partial protection in a mouse peanut allergy model [40].

*Lactobacillus* also had an inhibitory effect on respiratory infection. The group was randomly divided into the placebo group (*n* = 142) who consumed post-pasteurized fermented milk products without LGG, and the LGG group (*n* = 139) who consumed 100 mL of milk fermented with LGG. Compared to the placebo group, the LGG group significantly reduced their risk of upper respiratory tract infections. In addition, the number of days of respiratory symptoms was also significantly reduced [11]. *L. reuteri* DSM17938 also showed similar effects as LGG. The group that received the strain (*n* = 168) and a placebo group (*n* = 168) were administered the treatment daily for 3 months, after which they were followed up without supplementation for 3 months. The bacteria-treated group had a significantly decreased frequency and duration of diarrhea and respiratory infections at both 3 and 6 months than the placebo group [41]. Another species of *Lactobacillus*, *L. casei* CRL 431, showed an improvement effect on pneumoniae respiratory infection. The strain reduced the time required for a normal immune response from 21 days to 7 days, had effective pathogen clearance, and significantly reduced lung damage [42]. *Lactobacillus* also alleviated chronic asthma in a mouse model. In mice orally administered *L. rhamnosus* NutRes1, basal lung resistance was significantly increased compared to the control group that was orally administered PBS [43]. This species also attenuates cigarette smoke-induced chronic obstructive pulmonary disease (COPD). In the COPD group, which was orally administered *L. rhamnosus* three times per week, the influx of inflammatory cells into the airways was inhibited compared to that in the COPD group that did not receive *L. rhamnosus*. It was involved with various immune responses [44].

A positive effect of *Lactobacillus* on neurological and psychiatric diseases has been reported. When mice were chronically treated with *L. rhamnosus* JB-1, region-dependent alterations in GABAB1b mRNA were induced in the brain with increases in cortical regions and concomitant reductions in expression in the hippocampus, amygdala, and locus coeruleus compared to the controls. This bacterial strain, importantly, reduced stress-induced corticosterone and anxiety- and depression-related behaviors [45]. Psychotropic-like properties were shown in rats and healthy human volunteers fed *L. helveticus* R0052 combined with another bacterial strain (*B. longum* R0715). An anxiolytic-like activity was demonstrated when probiotics were administered daily for 2 weeks to anti-anxiety agents in a selection rat model. In addition, when these probiotics were administered to healthy clinical trial volunteers for 30 days, psychological distress was alleviated with statistically significant differences compared to baseline in the Hopkins symptom checklist, hospital anxiety and depression scale, and coping checklist [46]. The Probiotic IRT5 containing *L. casei*, *L. acidophilus*, and *L. reuteri*, which are known to have a positive effect against various diseases, reduced the incidence of experimental autoimmune encephalomyelitis (EAE). IRT5 also contained *B. bifidum* and *S. thermophilus*. Treatment with IRT5 before disease induction significantly inhibited the occurrence of EAE, and treatment with IRT5 for ongoing EAE delayed disease onset [47]. The probiotics also exhibited a protective effect against experimental autoimmune myasthenia gravis (EAMG). Oral administration of IRT5 probiotics five times per week significantly reduced symptoms of EAMG, such as weight loss, body tremors, and grip strength [48]. Another probiotic including *L. acidophilus* Rosell-11 affected arabinitol levels and behavior in autistic children. Autistic children who received the probiotic twice a day for 2 months had improved concentration and a better ability to follow instructions. In addition, the ratio of D-arabinitol to D-/L-arabinitol, which was high in autistic children, decreased in the urine [49].

*Lactobacillus* is also known for its inhibitory effect on genitourinary infections. The oral and intravaginal administration of *L. johnsonii* HY7042 to mice, induced to have vaginosis with *Gardnerella vaginalis*, had an inhibited myeloperoxidase activity in vaginal tissues, and a reduced population of *G. vaginalis* [12]. A positive effect of *Lactobacillus* on vaginosis was also observed in human experiments. Patients receiving *L. rhamnosus* BMX 54 (*n* = 125) had a significantly lower recurrence rate than subjects receiving only antibiotic treatment (*n* = 125). In addition, patients who received continuous supplementation during follow-up showed a significant decrease in pH compared to other subjects [50]. *L. crispatus* CTV-05 was effective at preventing urinary tract infection (UTI). After antibiotic treatment, the recurrence rate of UTIs was reduced from 27% in patients receiving placebo to 15% in patients receiving CTV-05. High levels of vaginal colonization of this strain throughout follow-up were associated with recurrent UTIs [51].

Probiotics have also been reported to have a disease-alleviating effect on metabolic syndromes such as diabetes and obesity. The incidence of diabetes was decreased in rats receiving *L. johnsonii* N6.2 daily [52]. When two bacterial strains, ADR-1 and ADR-3, of *L. rhamnosus* were orally administered to patients with type 2 diabetes mellitus (T2DM), significant reductions in HbA1c and serum cholesterol were observed in the ADR-1 group (*n* = 22) compared to the placebo group (*n* = 22). There was no significant difference in HbA1c serum levels in the heat-killed ADR-3 intake group (*n* = 24), however, the systolic and mean blood pressures were significantly decreased after 6 months of treatment with probiotics [53]. Metabolic syndrome is closely related to obesity. Mice receiving *L. gasseri* BNR17 along with a high-sucrose-diet had reduced body weight and reduced white adipose tissue weight. In addition, the mRNA level of fatty acid oxidation-related genes was significantly higher and that of fatty acid synthesis-related genes was lower than that of the high-sucrose-diet group [54]. Some species of *Lactobacillus* reduce the risk factors associated with cardiovascular disease. The serum total cholesterol, low-density lipoprotein cholesterol content levels, and atherosclerosis index were significantly decreased in rats fed *L. plantarum* DMDL9010. In addition, when morphological and pathological changes in the liver were observed, it was found to have a protective effect against hepatocellular steatosis. Decreasing of hepatic cholesterol and triglyceride levels and increasing of fecal excretion of bile acids were also observed [55].

A positive therapeutic effect of *Lactobacillus* is also observed in various oral diseases. The effect of *L. reuteri* ATCC 55730 and ATCC PTA5289 was confirmed in the group that received chewing gum containing probiotics and one placebo gum (*n* = 15), the group that received two gums containing each strain (*n* = 14), and the group that received two placebo gums (*n* = 13) for 2 weeks. Bleeding on probing (BOP) and gingival crevicular fluid were significantly improved in the two groups that received chewing gums containing probiotics compared to the placebo group [56].

*Lactobacillus* is known to suppress diseases through immune responses to various autoimmune diseases. In rats fed *L. casei*, collagen-induced arthritis was suppressed, and cartilage tissue destruction, paw swelling, and lymphocyte infiltration were reduced [61]. In mice, *L. fermentum* CECT5716 decreased the activity of lupus disease, blood pressure, cardiac and renal hypertrophy, and splenomegaly. This strain also reduced elevated T, B, regulatory T cells, T helper cells in mesenteric lymph nodes, and plasma lipopolysaccharide (LPS) levels [62]. Mice fed *L. paracasei* 1602 and *L. reuteri* 6798 showed reduced intestinal inflammation in IL-10-deficient mice compared with animals co-colonized with *Helicobacter hepaticus* at similar levels. In addition, proinflammatory colonic cytokine levels were reduced in the *Lactobacillus* spp. treated group [63].

*Lactobacillus* has been suggested to have a therapeutic effect against osteoporosis and tumors and to act as a vaccine adjuvant. In *L. acidophilus*-administrated ovariectomized mice, the bone mineral density and heterogeneity were increased and both the trabecular and cortical bone microstructure were enhanced [64]. Cell-bound EPS of *L. plantarum* 70810 was tested in vitro. The proliferation of HepG-2, BGC-823, and especially HT-29 tumor cells was significantly inhibited [65]. In a randomized double-blind placebo-controlled trial, LGG showed a protective titer of 1.84, a 95% confidence interval 1.04–3.22, and a *p* value of 0.048 compared to the control group 28 days after vaccination [66].

## 4. Effect of *Lactobacillus* in Liver Disease

The use of *Lactobacillus* as probiotics for liver diseases, NAFLD or ALD, are summarized in Table 2.

### 4.1. Non-Alcoholic Fatty Liver Disease

Non-alcohol fatty liver diseases (NAFLD) is the most common cause of chronic liver disease, however no definite treatment has been described thus far [96]. Probiotics have been proven in many studies to have anti-obesity effects. Based on the fact that many NAFLD patients are obese or overweight [97], it has been suggested that probiotics could be a new treatment for NAFLD [98]. The intake of *Lactobacillus*, the most commonly used probiotic, not only suppresses obesity caused by HFD, but also improves inflammation and regulation of the intestinal microbiome and increases the protective effect on the intestine [75,76].

The occurrence of NAFLD is associated with a disorder in liver lipid metabolism. Changes in lipid metabolism mainly cause fatty acid accumulation [99], and increase triglyceride accumulation in the liver [100]. Treatment with *L. plantarum* CQPC03 restored liver function and oxidative stress in mice and reduced fat accumulation in the liver. In addition to regulating lipid metabolism in the liver, it also alleviates inflammation by increasing the levels of interleukins IL-10 and IL-4, and decreasing the levels of proinflammatory factors, including IL-6, IL-1β, TNF-α, and IFN-γ [77]. Additionally, endotoxin levels and proinflammatory cytokines were significantly reduced, and the microbiota was controlled in the colon [78]. In mice fed a high-fat diet (HFD), the *Lactobacillus* strain can inhibit liver HMG-CoA reductase and make ferulic acid, which can promote the excretion of acidic sterol, showing that it is effective against NAFLD [101,102]. In addition, studies have shown that increased cholesterol accumulation contributes to liver damage, worsening NAFLD [103]. The group of mice treated with LGG upregulates the expression of cholesterol 7α-hydroxylase (CYP7A1), low density lipoprotein receptor (LDLR), and liver X receptor (LXR) genes, but downregulates the expression of small heterodimer partner (SHP) and farnesoid X receptor (FXR) [79]. These results indicate that it can inhibit cholesterol absorption and promote cholesterol transport. *Lactobacillus* suggests that bile salt hydrolase activity is a potential mechanism through which probiotics decrease cholesterol [104].

Over the past decade, many clinical trials have been performed to study the therapeutic effects of probiotics in NAFLD patients. When LGG was administered to obese children with NAFLD for 8 weeks, BMI decreased and alanine transaminase (ALT), TNF-α, and antipeptidoglycan-polysaccharide antibodies were significantly reduced [91]. In a study in which *L. acidophilus* was administered three times daily to adult NAFLD patients for 1 month, aminotransferase (AST) and ALT were significantly decreased. This showed that *Lactobacillus* helped improve the inflammatory condition of the patient’s liver [92]. As a result of a study in which *L. reuteri* and inulin were administered to NASH patients for 3 months, body weight, waist circumference, and BMI were reduced, and liver inflammation was improved [93]. The overall results indicate that *Lactobacillus* may be a promising therapeutic strategy for NAFLD (Table 2).

### 4.2. Alcoholic Liver Disease

Worldwide, alcoholic liver disease has the highest mortality rate among liver-related diseases. Alcoholic liver disease (ALD) is caused by bacterial translocation and LPS release due to intestinal barrier dysfunction, and intestinal-derived LPS plays a key role in increased liver inflammation and hepatic steatosis [105,106]. Probiotics alter the composition of the gut microbiota, reducing endotoxemia, bacterial translocation, dysbiosis, and consequently the development of ALD [107].

Several studies have found that animal models of ALD and ALD patients have abnormally very high LPS levels and increased intestinal permeability to alcohol-induced endotoxin [108]. In an animal model of chronic ALD, a *Lactobacillus* mixture (*L. acidophilus* KLDS1 and *L. plantarum* KLDS1.0344) decreased the serum LPS and improved the intestinal tight junction. It also inhibited inflammation, lipid accumulation, and oxidative stress through the gut−liver axis by regulating TLR4/NF-kB [80]. The effects of *Lactobacillus* on the gut−liver axis in ALD were evaluated by administering probiotics (*L. rhamnosus* R0011 and *L. acidophilus* R0052) together to ALD-induced mice inducted by intraperitoneal injection of ethanol and LPS. Consequently, *Lactobacillus* regulated alcohol-induced TLR4 overexpression and decreased proinflammatory cytokines (TNF-α, IL-1β, and IL-6) and ALT levels [81]. LGG treatment reduced alcohol-induced liver inflammation by attenuating TNF-α production through the inhibition of TLR4- and TLR5-mediated endotoxin activation [82]. In the ALD animal model modeled by Gao-binge, the group administered with *L. reuteri* had alleviated inflammatory cell infiltration and lipid accumulation. In addition, AST, ALT, triglyceride (TG), and total cholesterol (TCH) levels were also decreased [83].

In human study, supplementation of patients with alcoholic liver injury with *L. casei* significantly decreased their serum TG and LDL-C. The amounts of *Lactobacillus* and *Bifidobacterium* were increased compared to those in the control group; consequently, gut microbiota disorders were controlled, and lipid metabolism was improved [94]. Many studies have shown that *Lactobacillus* improves ALD. However, more research is needed to explain the mechanism behind this.

### 4.3. Liver Fibrosis and Cirrhosis

Liver fibrosis occurs when the tissue in the liver is damaged or inflamed and does not work properly [109]. There are many causes of liver fibrosis, including hepatitis virus, alcohol consumption, and bile acid accumulation [110]. When liver fibrosis progresses throughout, it becomes liver cirrhosis.

Hepatic accumulation of bile acids (BAs) plays a key role in the pathogenesis of cholestasis-induced liver injury, and an excess of cytotoxic BAs in the liver can lead to liver fibrosis and cirrhosis [111]. LGG supplementation increases the intestinal FXR-FGF-15 signaling pathway-mediated inhibition of BA de novo synthesis to reduce hepatic BA and enhance BA excretion to prevent excessive bile acid induced liver injury and fibrosis in mice [84]. In another study, ccl4 injection with *L. salivarius* LI01 reduced liver inflammatory responses, including TNF-α, INOS-2, TGF-β, IL-17A, and IL-6, and decreased AST and ALT. As a result, it relieved hepatocellular damage and showed anti-inflammatory effects [86]. In addition, *Lactobacillus* reduced the expression of fibrosis-related genes (Timp1, TGF-β1, Col1α1, and Acta2), and intestinal barrier function was also improved by enhancing the expression of tight junction protein [85,86]. In patients, LGG is tolerated in liver cirrhosis patients and is involved in the reduction of endotoxin and dysbiosis [95]. In another study, in the case of cirrhosis patients administered a capsule of probiotics containing *L. acidophilus* and *L. bulgaricus*, no significant effect was found, but ammonia levels were reduced in patients with ammonia levels above the normal baseline [112]. The effect of *Lactobacillus* needs further investigation in a larger cohort.

### 4.4. Hepatocellular Carcinoma

Hepatocellular carcinoma (HCC) is a hepatocyte cancer and mainly occurs in patients with viral infection, alcohol-induced cirrhosis, or non-alcohol associated steatohepatitis (NASH) [113,114]. Probiotics are food supplements containing microbes for human consumption and may be applied as biotherapeutic agents because they have a beneficial effect on health due to desirable changes in the intestinal microbial balance. While general cancer treatment methods have many side effects, biological treatments such as probiotic intake do not have side effects [115].

In a mouse model of hepatocellular carcinoma, the administration of LGG has been shown to reduce tumor progression [87]. In an animal study using azoxymethane (AOM), which is a carcinogen for mouse colon and liver cells, *L. acidophilus* downregulated the oncogenes (Bcl-w and KRAS) and up-regulated PTEN, a tumor suppressor gene, compared to the control group. Therefore, it helped control cancer progression [88]. *L. acidophilus* LA14 reduces liver damage caused by D-GaLN (D-galactosamine) by alleviating upregulated ROCK2 (which promotes hepatocellular carcinoma), FBLM1 (which promotes cancer progression), and COL12A1, a collagen type XII α1 chain. This means that LA14 prevents hepatocellular carcinoma during liver injury [89]. When hepatocellular carcinoma-induced rats treated with diethylnitrosamine (DEN) and gamma radiation (IR) were treated with EPS produced by *L. acidophilus* ATCC 4356, serum ALT and γ-GT activities were alleviated. Additionally, MDA, IL-17, and TGF-β1 were also ameliorated, and they showed a preventive effect on HCC through the regulation of the inflammation-related TLR2/STAT-3/P38-MAPK pathway [90]. In vitro, *Lactobacillus* strains (*L. acidophilus* HM1, *L. buchneri* FD2, and *L. fermentum* HM3) also showed a strong inhibitory activity against liver cancer HepG2 cells and showed selectivity for the apoptosis of cancer cells compared to normal cells [116].

The cell wall components of *L. acidophilus* and *L. casei* act as anticancer substances [117], and *L. plantarum* 70810 EPS prevents the proliferation of hepatocellular carcinoma cell lines [65]. Accordingly, *Lactobacillus* may be a prospective probiotic for the prevention and treatment of HCC, but further clinical studies are needed.

## 5. Perspective

Still, there is no definitive treatment method for liver disease. Many studies are being conducted to improve liver disease by using *Lactobacillus* as well as several probiotics to reduce hepatic steatosis and inflammation and control the microbiome. More research is needed on how probiotics improve liver disease.

Biological treatments, such as probiotics, currently have no side effects and are emerging as microbial drug candidates. However, because each gut microbiome is different, it is important to elucidate the mechanisms by which probiotics affect liver disease, and further studies are needed.

Probiotics can be applied as biotherapeutics because they have beneficial effects on health through changes in microbial balance. For probiotics, lactic acid bacteria such as *Bifidobacterium* and *Lactobacillus* are widely used as microorganisms. Among them, the genus *Lactobacillus* is one of the most studied microbial genera, and many of them have completed whole genome sequence analysis and are still in progress. There are many animal studies that *Lactobacillus* improves liver disease. Based on these, many clinical trials are currently underway, and further studies in a larger cohort study needed (Table 3).

The nomenclature of the genus *Lactobacillus* has been recently changed, therefore, the 315 species of *Lactobacillus*, as mentioned above, were divided into different 23 genera [118]. *Lactobacillus* mentioned in this review followed the nomenclature before it changed, therefore the correct nomenclature of each species is summarized in Table 4. The species of *Lactobacillus* not mentioned in Table 4 are to keep the original name.

## 6. Conclusions

Preclinical clinical and clinical studies have provided evidence for the various disease-alleviation effects of probiotics. *Lactobacillus*, particularly, has improved effects on a wide range of diseases with mechanisms. In this review, the disease improvement mechanisms and various disease suppression effects of *Lactobacillus*, especially on liver disease, were discussed in detail. As summarized in Figure 1, *Lactobacillus* positively contributes to various diseases with antimicrobial activity, microbiota modulation, antitumor activity, and immunomodulatory effects. However, not all mechanisms for all diseases have been elucidated yet. Even in the same species, the difference in disease alleviation effect depending on the strain is one of the causes of this difficulty. In addition, it is considered that more developed technologies are needed to elucidate the mechanisms of bacteria used for all probiotics, as well as *Lactobacillus*. As mentioned above, probiotics are emerging as microbial medicine candidates. It could be more helpful, especially, for research where the medicines have not yet been developed, such as liver disease. Therefore, understanding the bacterial strains and the development of technologies with cohort studies will facilitate the use of *Lactobacillus* in the treatment of various diseases.

## Figures and Tables

**Figure 1 microorganisms-10-00288-f001:**
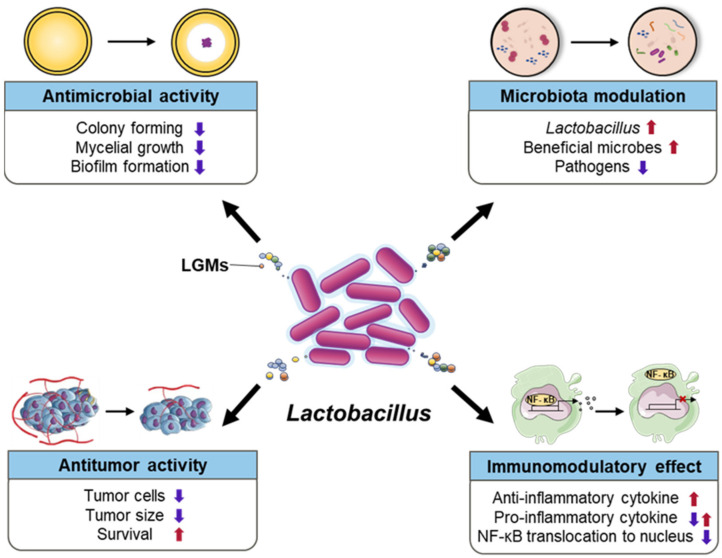
Various disease control mechanisms of the genus *Lactobacillus*. LGMs indicate *Lactobacillus* generated molecules.

**Table 1 microorganisms-10-00288-t001:** Use of *Lactobacillus* as probiotics for various diseases.

Classification of Diseases	Disease or Pathogen	Subject	Probiotics	Outcomes	Ref.
Gastrointestinal diseases	*C. difficile*	Human	*L. paracasei* F19	Reduced the population of *C. difficile*, which can cause diarrhea and enteritis.	[32]
	Acute watery diarrhea	Human	*L. rhamnosus* GG	Effective in reducing the frequency and duration of diarrhea in patients with different concentrations of the bacterium (10^10^ and 10^12^).	[33]
	Ulcerative colitis	Human	*L. rhamnosus* GG	Effective and safe for maintaining remission in patients with ulcerative colitis.	[34]
	Functional bowel disorders	Human	*L. acidophilus* NCFM (combined with another bacterium)	Improved symptoms of bloating.	[35]
	Colitis	Mouse	*L. acidophilus*, *L. bulgaricus*, *L. casei*, *L. plantarum* (combined with other bacteria)	Improved dextran sulfate sodium induced colitis.	[36]
Allergy	Allergic sensitization	Mouse	VSL#3	Reduced systemic and local anaphylactic symptoms by oral challenge with the sensitizing allergen Shrimp Tropomyosin.	[37]
	Atopic dermatitis	Human	*L. salivarius* LS01	Improved in scoring atopic dermatitis and itch values from baseline.	[38]
	Perennial allergic rhinitis	Human	*L. acidophilus* L-92	Alleviated the symptoms.	[39]
	Allergic rhinitis	Human	*L. paracasei* KW3110	Reduction of nasal symptoms and the serum level of eosinophil cationic protein and improvement of quality-of-life scores when pollen scattering was low.	[13]
	Food allergy (peanut)	Mouse	*L. salivarius* HMI001, *L. casei* Shirota	Partial protection in a mouse peanut allergy model.	[40]
Respiratory diseases	Gastrointestinal and respiratory tract infections	Human	*L. rhamnosus* GG	Reduced risk of upper respiratory tract infections, respiratory tract infections, and number of days with respiratory symptoms.	[11]
	Diarrhea and respiratory tract infection	Human	*L. reuteri* DSM 17938	Reduced the frequency and duration of diarrhea and respiratory infections, and consequently reduced costs for the community.	[41]
	Pneumococcal respiratory infection	Mouse	*L. casei* CRL 431	Accelerated the recovery of the innate immune system.	[42]
	Chronic asthma	Mouse	*L. rhamnosus* NutRes1	Reduced lung resistance in a mouse model of chronic asthma to a similar extent to budesonide treatment.	[43]
	Chronic obstructive pulmonary disease	Mouse	*L. rhamnosus*	Regulates pro- and anti-inflammatory cytokines balance in human bronchial epithelial cells and alleviates pulmonary inflammatory responses.	[44]
Neurological and psychiatric diseases	Neurological and psychiatric diseases	Mouse	*L. rhamnosus* JB-1	Reduced stress-induced corticosterone and anxiety- and depression-related behaviors.	[45]
	Neurological and psychiatric diseases	Human/Rat	*L. helveticus* R0052 (combined with another bacterium)	Anxiolytic-like activity in rats, beneficial psychological effects in healthy humans.	[46]
	Neurological and psychiatric diseases	Mouse	*L. casei*, *L. acidophilus*, *L. reuteri* (combined with other bacteria) (IRT5)	Suppressed experimental autoimmune encephalomyelitis.	[47]
	Autoimmune myasthenia gravis	Rat	IRT5	Prevented the development of experimental autoimmune myasthenia gravis.	[48]
	Autism spectrum disorder	Human	*L. acidophilus* Rosell-11	Reduced D-arabinitol level and D-/L-arabinitol ratio in urine and improved concentration and carrying out orders.	[49]
Genito-Urinary tract infections	Bacterial vaginosis	Mouse	*L. johnsonii* HY7042	Inhibited myeloperoxidase activity in vaginal tissue and reduced viable numbers of *Gardnerella vaginalis*.	[12]
	Bacterial vaginosis	Human	*L. rhamnosus* BMX 54	Reduced recurrence rate and reduced pH.	[50]
	Urinary tract infections	Human	*L. crispatus* CTV-05	Reduced recurrence.	[51]
Metabolic syndrome	Type 1 diabetes	Rat	*L johnsonii* N6.2	Mitigated the development of type 1 diabetes.	[52]
	Type 2 diabetes mellitus	Human	*L. reuteri* ADR-1, *L. reuteri* ADR-3	Beneficial effect on patients.	[53]
	Obesity	Mouse	*L. gasseri* BNR17	Decreased leptin and insulin levels in serum and showed anti-obesity effects.	[54]
	Cardiovascular disease	Rat	*L. plantarum* DMDL 9010	Decreased serum and total liver cholesterol and triglyceride and enhanced fecal excretion of bile acids.	[55]
Oral diseases	Gingivitis	Human	*L. reuteri* ATCC 55730, *L. reuteri* ATCC PTA 5289	Decreased bleeding on probing and gingival crevicular fluid during chewing gums containing probiotics.	[56]
	Periodontitis	Human	*L. reuteri* DSM 17938, *L. reuteri* ATCC PTA 5289	Improved clinical parameters and reduced abundance of pathogenic bacterium.	[57]
	Dental caries	Human	*L. rhamnosus* GG	Reduced the risk of caries and lowered mutans *Streptococcus* counts.	[58]
	Halitosis	Human	*L. salivarius* WB21	Decreased an organoleptic test and BOP.	[59]
	Oral candidiasis	In vitro	*L. fermentum* 20.4, *L. paracasei* 28.4, *L. rhamnosus* 5.2	Inhibited biofilms of *Candida albicans.*	[60]
Autoimmune diseases	Rheumatoid arthritis	Rat	*L. casei*	Suppressed collagen-induced arthritis and reduced destruction of cartilage tissue, paw swelling, and lymphocyte infiltration.	[61]
	Systemic lupus erythematosus	Mouse	*L. fermentum* CECT5716	Reduced activity of lupus disease.	[62]
	Inflammatory bowel disease	Mouse	*L. paracasei* 1602, *L. reuteri* 6798	Reduced intestinal inflammation *Helicobacter hepaticus*-challenged IL-10-deficient mice.	[63]
Others	Osteoporosis	Mouse	*L. acidophilus* ATCC 4356	Increased bones’ mineral density and heterogeneity and enhanced trabecular and cortical bone microstructure.	[64]
	Tumor cells	In vitro	*L. plantarum* 70810	Inhibited the proliferation of tumor cells.	[65]
	Vaccine adjuvant	Human	*L. rhamnosus* GG	Had a protective titer 28-day-after vaccination.	[66]

**Table 2 microorganisms-10-00288-t002:** Use of *Lactobacillus* as probiotics for liver diseases.

Subject	Disease	Treatment	Main Effect	Ref.
Animal	NAFLD	*L. gasseri* SBT2055	(↓): body weight, pro-inflammatory (CCL2, CCR2, TNF), LPS (↑): intestinal barrier function, permeability	[75]
		*L. rhamnosus* GG	(↓): ALT, liver inflammation (IL-8R, IL-1β) and steatosis, LPS, TNF-α (↑): chREBP, FAS, ACC1 total numbers of the distal small intestinal microbiota, major tight junction proteins (occludin and claudin-1)	[76]
		*L. plantarum* CQPC03	(↓): hepatic tissue damage, hepatic triglyceride, total cholesterol, IL-6, IL-1β, TNF-α, interferon-γ (↑): HDL-C, IL-4, IL-10, SOD, GSH-Px, lipoprotein lipase	[77]
		*L. plantarum* NCU116	(↓): liver enzymes, bilirubin, IL-6, TNF-α, IL-10, oxidative stress, fat accumulation in the liver, lipogenesis, LPS (↑): fatty acid oxidation	[78]
		*L. casei* pWQH01, *L. plantarum* AR113	(↓): Body weight, total cholesterol, atherogenic index, small heterodimer partner, farnesoid X receptor (↑): cholesterol 7α-hydroxylase, liver X receptor, lipoprotein receptor	[79]
	ALD	*L. acidophilus* KLDS1, *L. plantarum* KLDS1.0344	(↓): liver enzymes, LPS, oxidative stress, inflammation, lipid accumulation (↑): intestinal epithelial permeability	[80]
		*L. rhamnosus* R0011, *L. acidophilus* R0052	(↓): TNF-α, IL-1β, IL-6, TLR4 expression, IL-10	[81]
		*L. rhamnosus* GG	(↓): TNF-α, CYP2E1, LPS, phosphorylation of p38 MAP kinase, nuclear factor erythroid 2-related factor 2 expression	[82]
		*L. reuteri*	(↓): liver enzymes, lipid accumulation, inflammation, LPS, (↑): ZO-1, linoleic acid, arachidonic acid	[83]
	Fibrosis	*L. rhamnosus* GG	(↓): Hepatic bile acid, liver inflammation, liver injury, hepatic cholesterol 7α-hydroxylase (↑): expression of serum and ileum fibroblast growth factor 15	[84]
		*L. paracasei*, *L. casei*	(↓): inflammation, TNF-α, TGF-β1, α-SMA proteins, Col1a1, Acta2, Timp1, TGF-β	[85]
	Cirrhosis	*L. salivarius* LI01	(↓): Serum endotoxin, bacterial translocations, TNF-α, IL-6, IL-17A, TLR2, TLR9, TLR5, liver enzymes (↑): Intestinal barrier	[86]
	HCC	*L. rhamnosus* GG	(↓): IL-17, recruitment of Th17 from gut to tumor sites, tumor progression (↑): IL-10, antitumor function	[87]
		*L. acidophilus*	(↓): oncogene, MiR-122, oncomir, tumor suppressor gene (↑): tumor suppressor gene	[88]
		*L. acidophilus* LA14	(↓): liver enzymes, bile acid, histological injury to the gut and liver, inflammatory cytokines	[89]
		*L. acidophilus* ATCC 4356	(↓): liver enzymes, IL-17, TGF-β1	[90]
Human	NAFLD	*L. rhamnosus* GG	(↓): BMI, ALT, TNF-α, alanine aminotransferase, antipeptidoglycan-polysaccharide antibodies	[91]
	NAFLD/NASH	*L. acidophilus*	(↓): liver enzymes, dyspepsia	[92]
	NASH	*L. reuteri* + inulin	(↓): Body weight, waist circumference, BMI	[93]
	ALD	*L. casei*	(↓): TG, LDL-C, liver enzymes, TNF-α, IL-1β, IL-6, intestinal flora imbalance (↑): Amount of *Lactobacillus* and *Bifidobacterium* in the intestinal flora, improve lipid metabolism	[94]
	Cirrhosis	*L. rhamnosus* GG	(↓): endotoxemia, dysbiosis, TNF-α	[95]

(↓) indicates a decrease in condition; (↑) indicates an increase in condition; NAFLD: non-alcoholic fatty liver disease; ALD: alcoholic liver disease; HCC: hepatocellular carcinoma; CCL2: chemokine ligand 2; CCR2: C-C chemokine receptor type 2; TNF: tumor necrosis factor; LPS: lipopolysaccharide; IL-8R: interleukin-8 receptor; IL: interleukin; chREBP: Carbohydrate response element binding protein; ACC1: acetyl-CoA carboxylase 1; LDL-C: low-density lipoprotein-cholesterol; SOD: superoxide dismutase; GSH-Px: glutathione peroxidase; ALT: alanine aminotransferase; TLR: toll-like receptor; CYP2E1: cytochrome P4502E1; TG: triglyceride; αSMA: α-smooth muscle actin; Col1a1: collagen type 1 alpha 1; Timp1: metallopeptidase inhibitor 1; TGF-β: transforming growth factor-beta.

**Table 3 microorganisms-10-00288-t003:** Clinical trials currently in progress with *Lactobacillus*.

Status	Disease	Study Title	Interventions	Identifier
Recruiting	NAFLD	Role of probiotics in Treatment of pediatric NAFLD patients by assessing with fibroscan	Dietary supplement: culturelle (*L. rhamnosus* GG) Other: placebo	NCT04671186
Unknown	NAFLD	Probiotics in the treatment of NAFLD	• Dietary supplement: probiotic *L. acidophilus* 10⁹, *B. lactis* 10⁹.	NCT02764047
Unknown	ALD	Effect of probiotics on gut-liver axis of alcoholic hepatitis	• Drug: Probiotics (Lacidofil^®^) 7 days of cultured *L. rhamnosus* R0011/*L. acidophilus* R0052 (120 mg/day) • Drug: Placebo	NCT02335632
Recruiting	ALD	Alcoholic liver disease and the gut microbiome	• Drug: VSL#3 112.5 Capsule A commercial probiotic mixture consisting of Four strains of *L.* (*L. casei*, *L. acidophilus*, *L. delbrueckii* subspecies *bulgaricus*, and *L. plantarum*), three strains of *Bifidobacterium*, and one strain of *Streptococcus*. • Other: Placebo	NCT05007470
Suspended	Acute alcoholic hepatitis	Novel therapies in moderately severe acute alcoholic hepatitis	Dietary supplement: *L. rhamnosus* GG Drug: placebo for probiotic	NCT01922895
Active, not recruiting	ALD fibrosis cirrhosis	Profermin^®^: prevention of progression in alcoholic liver disease by modulating dysbiotic microbiota	• Profermin Plus^®^ Based on fermented oats, *L. plantarum* 299v, lecithin and barley malt.	NCT03863730
Completed	Cirrhosis	Influence of probiotics on infections in cirrhosis	• Wonclove-849 (*L. brevis* W63, *L. salivarius* W24, *L. casei* W56, *L. acidophilus* W37, *Lactococcus lactis* W19, *Lactococcus lactis*, *B. bifidum* W23, *B. lactis* W52)	NCT01607528
Not yet recruiting	cirrhosis hepatocellular carcinoma	Probiotics in the Prevention of Hepatocellular carcinoma in cirrhosis	• Probiotics contains *L. casei*, *L. plantarum*, *Streptococcus faecalis*, and *B. brevis*	NCT03853928
Completed	Fibrosis Cirrhosis hepatocellular carcinoma	Influence of probiotics administration before liver resection in liver disease	• Active substance mixture of lactic 10% *B. lactis* LA 303, 10% *L. acidophilus* LA 201, LA 40% *L. plantarum* 301, 20% *L. salivarius* LA 302, LA 20% *B. lactis* 304 Dosage: 10 × 10^9^ probiotic/capsule	NCT02021253

NAFLD, non-alcoholic fatty liver disease; ALD, alcoholic liver disease.

**Table 4 microorganisms-10-00288-t004:** Correct nomenclature of *Lactobacillus* spp.

Old Nomenclature	Correct Nomenclature	Reference
*L. casei*	*Lacticaseibacillus casei*	[118]
*L. fermentum*	*Limosilactobacillus fermentum*	
*L. reuteri*	*Limosilactobacillus reuteri*	
*L. rhamnosus*	*Lacticaseibacillus rhamnosus*	
*L. plantarum*	*Lactiplantibacillus plantarum*	
*L. paracasei*	*Lacticaseibacillus paracasei*	
*L. salivarius*	*Ligilactobacillus salivarius*	

## Data Availability

Data are contained within the article.

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
