# Peer review of "The Lactobacillus as a Probiotic: Focusing on Liver Diseases"

_microorganisms, 2022, doi:10.3390/microorganisms10020288_

Round 1
Reviewer 1 Report
Dear Authors,
Overall, the manuscript is very well written. And it was interesting to see how the Lactobacillus was able to relieve the symptoms of liver diseases through having antimicrobial activity, microbiota modulation, antitumor activity, as well as immunomodulatory effect. Also, the manuscript is well organized and summarized the use of Lactobacillus, mechanisms of action, immunoregulation such as the production of cytokines and toll-like receptor pathways, and the current clinical trials. For these reasons, I do not have any comments for the current manuscript, and it was my pleasure to review this manuscript.
Author Response
we would like to thank the Reviewer 1 for his/her comments, which helped us to improve this manuscript.
Reviewer 2 Report
Dear Authors,
Please refer to the attached Word file with my comments. Thank you.
JS

Author Response
- Comment 1: I have only one main comment for the authors to consider and respond in the best possible way. The authors should be aware that the Lactobacillus nomenclature has totally changed recently (viz. Zheng et al., 2020, Int. J. Syst. Evol. Microbiol. 70, 2782-2858). The old genus Lactobacillus with 315 species does not exist anymore! It was split in 23 novel genera with emended names. Only one genomic group of dairy species retains Lactobacillus. Most other Lactobacillus spp. reported in Tables 1 and 2 and throughout the text with their old names must currently be revised according to Zhang et al. This change is laborious and generates editing complexity, but it is essential for the scientific credibility of this review. To assist the authors, alternatively, I suggest to include a remark in the abstract, start the Introduction with a short relevant paragraph to cite the above important IJSEM article. Then list all cited Lactobacillus spp. with their emended names, accompanied by a statement that thereafter the old nomenclature is used for convenience, given that most (if not all) studies cited in the References section have been published by using the old Lactobacillus names.
- Response 1: Thanks for this comment. Referring to the comment, we summarized the content related to the reclassification of Lactobacillus in abstract L19, main text L121 and perspective L477-L481. In addition, Table 4 has been added along with citation of the paper on the reclassification of Lactobacillus. Table 4 shows the nomenclature before reclassification used in this paper and the revised correct nomenclature.
Table 4. Correct nomenclature of Lactobacillus spp.
|
Old nomenclature |
Correct nomenclature |
Reference |
|
L. casei |
Lacticaseibacillus casei |
[119] |
|
L. fermentum |
Limosilactobacillus fermentum |
|
|
L. reuteri |
Limosilactobacillus reuteri |
|
|
L. rhamnosus |
Lacticaseibacillus rhamnosus |
|
|
L. plantarum |
Lactiplantibacillus plantarum |
|
|
L. paracasei |
Lacticaseibacillus paracasei |
|
|
L. salivarius |
Ligilactobacillus salivarius |
- Comment 2: L46: Suggest change microflora to microbiota.
- Response 2: We modified microflora to microbiota.
- Comment 3: L55: Delete “However”. Start the sentence ‘The gut microbiome………practice, however,…..
- Response 3: Thanks for reviewer’s consideration. We deleted “However” and changed the sentence.
- Comment 4: L65: Lactobacillus and Saccharomyces genera
- Response 4: We changed genus to genera.
- Comment 5: L72: Correct stains as strains
- Response 5: Stains were corrected for strains
- Comment 6: L79: ….. in the gut of high-fat diet…….
- Response 6: We modified ‘in the gut high-fat diet’ to ‘in the gut of high-fat diet.’
- Comment 7: L98 to L111: I think something is wrong with the citing numbers used in this part. Are you sure that refs [23, 24] in L99 are the correct ones cited? In L104, I think the correct citation is 24, not 25. Also, citation 25 about skin seems needed in the bracket {27, 28], L111. Are 27 and 28 okay here? Please check and amend the numbers, if necessary.
- Response 7: We checked all the papers commented on by reveiwer and re-quoted them as correct citations.
[23, 24] papers previously cited in L99 were deleted and cited again with the correct papers.
Previously cited paper
- Sokol, H.; Pigneur, B.; Watterlot, L.; Lakhdari, O.; Bermúdez-Humarán, L.G.; Gratadoux, J.-J.; Blugeon, S.; Bridonneau, C.; Furet, J.-P.; Corthier, G. Faecalibacterium prausnitzii is an anti-inflammatory commensal bacterium identified by gut microbiota analysis of Crohn disease patients. Proceedings of the National Academy of Sciences 2008, 105, 16731-16736.
- Singh, R.; Van Nood, E.; Nieuwdorp, M.; Van Dam, B.; Ten Berge, I.; Geerlings, S.; Bemelman, F. Donor feces infusion for eradication of extended spectrum beta-lactamase producing Escherichia coli in a patient with end stage renal disease. Clinical Microbiology and Infection 2014, 20, O977-O978.
Corrected paper
- Singhi, S.C.; Kumar, S. Probiotics in critically ill children. F1000Research 2016, 5.
- Dallal, M.S.; Davoodabadi, A.; Abdi, M.; Hajiabdolbaghi, M.; Yazdi, M.S.; Douraghi, M.; Bafghi, S.T. Inhibitory effect of Lactobacillus plantarum and Lb. fermentum isolated from the faeces of healthy infants against nonfermentative bacteria causing nosocomial infections. New Microbes and New Infections 2017, 15, 9-13.
In L104, we also corrected the citation.
Previously cited paper
- Krutmann, J. Pre-and probiotics for human skin. Clinics in plastic surgery 2012, 39, 59-64.
Corrected paper
- Caballero, S.; Carter, R.; Ke, X.; Sušac, B.; Leiner, I.M.; Kim, G.J.; Miller, L.; Ling, L.; Manova, K.; Pamer, E.G. Distinct but spatially overlapping intestinal niches for vancomycin-resistant Enterococcus faecium and carbapenem-resistant Klebsiella pneumoniae. PLoS Pathogens 2015, 11, e1005132.
In L111, citations on skin were also corrected to be right citations.
Previously cited paper
- Borgeraas, H.; Johnson, L.; Skattebu, J.; Hertel, J.; Hjelmesaeth, J. Effects of probiotics on body weight, body mass index, fat mass and fat percentage in subjects with overweight or obesity: a systematic review and meta‐analysis of randomized controlled trials. Obesity reviews 2018, 19, 219-232.
- Monteiro-Sepulveda, M.; Touch, S.; Mendes-Sá, C.; André, S.; Poitou, C.; Allatif, O.; Cotillard, A.; Fohrer-Ting, H.; Hubert, E.-L.; Remark, R. Jejunal T cell inflammation in human obesity correlates with decreased enterocyte insulin signaling. Cell metabolism 2015, 22, 113-124.
Corrected paper
- Bowe, W.P.; Logan, A.C. Acne vulgaris, probiotics and the gut-brain-skin axis-back to the future? Gut Pathogens 2011, 3, 1-11.
- Muizzuddin, N.; Maher, W.; Sullivan, M.; Schnittger, S.; Mammone, T. Physiological effect of a probiotic on skin. Journal of Cosmetic Science 2012, 63, 385-395.
- Comment 8: L116: ……can be able…..
- Response 8: We changed ‘can able’ to ‘can be able’.
- Comment 9: L123: …..which belong to Lactobacillus are…..
- Response 9: The abbreviated Lactobacillus was rewritten as the full name.
- Comment 10: L139: L. rhamnosus with ., not comma
- Response 10: We changed the comma (,), incorrectly written in front of rhamnosus, to full-stop (.).
- Comment 11: L157: Correct salicarius
- Response 11: We corrected species name misspelled as salicarius to salivarius.
- Comment 12: L166: virginal??? Do you mean vaginal?
- Response 12: We modified virginal to vaginal.
- Comment 13: L189-L191: This is an exact text repetition of previous L160-162.
- Response 13: Because L189-L191 contains the contents of the effect of Lactobacillus as a probiotic, we modified the L160-L162 to focus on the mechanism.
- Comment 14: L243-244: The sentence “This species also……….disease (COPD)” is written twice (viz. L242-243 above). Please delete the second sentence. Section 4.1, starting in L330: From this point, the text becomes problematic in following the order number of the cited references because the refs 96, 97 and 98 in L332-335 are cited before the refs 75, 76 in L337. The same editing problem continues in next paragraphs, just because all references in order from 75 to 95 are cited for the first time in Table 2. So, to fix this problem and the readers confusion, Table 2 must be moved under the heading section 4, just before starting section 4.1, accompanied by an introductory sentence ‘The use of Lactobacillus as probiotics for liver, NAFLD or ALD, diseases are summarized in Table 2’.
- Response 14: L243-244: We deleted one of the duplicated sentences.
Section 4.1, starting in L330: We rearranged Table 2 under the heading section of 4, and added a description ‘The use of Lactobacillus as probiotics for liver, NAFLD or ALD, diseases are summa-rized in Table 2.’ of Table 2 just before it.
- Comment 15: L366: I think the correct citation here is 83, not 93.
- Response 15: We cited 93 in L366 because the paper is related to nonalcohol associated steatohepatitis, and cited 83 in the alcohol liver disease part..
- Comment 16: L469: Italicize Bifidobacterium
- Response 16: We modified Bifidobacterium in italics.
- Comment 17: Table 3, in horizontal box 4: Strains of Lactobacillus (L. casei, L. acidophilus….)
- Response 17: We modified strains of L. (L. casei, …) to strains of Lactobacillus (L. casei,…)
- Comment 18: L478: Delete the first ‘clinical’, the same word written twice.
- Response 18: We delete the first ‘clinical’ of the double written ‘clinical’.
- Comment 19: Please check all references for editing style. For instance, in refs 1, 2, 8 and others all the main words in a journal name should have the first letter capitalized, e.g. International Journal of Food microbiology, in ref 1.
- Response 19: The first letter of the journal name changed to capital letters.